# Shading Treatment Reduces Grape Sugar Content by Suppressing Photosynthesis-Antenna Protein Pathway Gene Expression in Grape Berries

**DOI:** 10.3390/ijms25095029

**Published:** 2024-05-05

**Authors:** Xintong Nan, Wenfang Li, Miao Shao, Zimeng Cui, Han Wang, Jiaxing Huo, Lizhen Chen, Baihong Chen, Zonghuan Ma

**Affiliations:** College of Horticulture, Gansu Agricultural University, Lanzhou 730070, China17662835503@163.com (J.H.);

**Keywords:** grape berries, cluster shading, sugar content, gene expression, photosynthesis

## Abstract

To explore the impact of shade treatment on grape berries, ‘Marselan’ grape berries were bagged under different light transmission rates (100% (CK), 75% (A), 50% (B), 25% (C), 0% (D)). It was observed that this treatment delayed the ripening of the grape berries. The individual weight of the grape berries, as well as the content of fructose, glucose, soluble sugars, and organic acids in the berries, was measured at 90, 100, and 125 days after flowering (DAF90, DAF100, DAF125). The results revealed that shading treatment reduced the sugar content in grape berries; the levels of fructose and glucose were higher in the CK treatment compared to the other treatments, and they increased with the duration of the shading treatment. Conversely, the sucrose content exhibited the opposite trend. Additionally, as the weight of the grape berries increased, the content of soluble solids and soluble sugars in the berries also increased, while the titratable acidity decreased. Furthermore, 16 differentially expressed genes (DEGs) were identified in the photosynthesis-antenna protein pathway from the transcriptome sequencing data. Correlation analysis revealed that the expression levels of genes VIT_08s0007g02190 (Lhcb4) and VIT_15s0024g00040 (Lhca3) were positively correlated with sugar content in the berries at DAF100, but negatively correlated at DAF125. qRT-PCR results confirmed the correlation analysis. This indicates that shading grape clusters inhibits the expression of genes in the photosynthesis-antenna protein pathway in the grape berries, leading to a decrease in sugar content. This finding contributes to a deeper understanding of the impact mechanisms of grape cluster shading on berry quality, providing important scientific grounds for improving grape berry quality.

## 1. Introduction

Grape (*Vitis vinifera* L.) is a woody vine belonging to the *Vitis* family. China has a long history of cultivating grapes, dating back to over 2000 years ago [1]. Grapes are classified into wine grapes and table grapes. Wine grapes are predominantly cultivated in Gansu Province, which plays a pivotal role in China’s grape industry [2]. About 80% of the world’s countries are exhibiting a growing trend of wine imports, and the demand for wine among people is increasing [3]. Grapes are rich in various nutrients, including carbohydrates, protein, and vitamins, providing essential nutrients that are crucial for human health [4]. The sugar content of grape berries is a crucial factor that influences the quality of grapes. Starch accumulation during berry development is closely linked to the soluble sugar content in grape berries [5,6]. The sugar components in grape berries include sucrose, glucose, fructose, sorbitol, xylose, erythritol, raffinose, and others [7]. The most abundant sugars in ripe grape berries are fructose and glucose, with a ratio close to one. Sucrose, on the other hand, accounts for more than 90% of the soluble sugar content in ripe grape berries [8,9]. The composition and proportion of sugar and acids (malic acid and citric acid) in grape berries not only directly affect the sweetness and taste of the berries but also serve as the fundamental raw materials for the synthesis of acids, pigments, amino acids, and other components [10,11]. These elements are closely related to the sweetness and nutritional value of the berries. Therefore, enhancing the quality of berries is of great significance for the advancement of the grape industry. With the improvement of people’s living standards, consumers are no longer satisfied with just the quantity of grapes and berries. They now pay more attention to product diversity, taste, nutritional value, and health benefits [12].

Previous studies have been conducted on the effect of shading treatments on the color and quality of wine in the northern foothills of the Xinjiang Tianshan Mountains. Shading treatment can increase the content of anthocyanins in wine and delay the yellowing process of wine during aging [13]. In addition, the total amount of anthocyanins in Norton grape skins is significantly higher than that in Cabernet Sauvignon grape skins during harvest, which is an important wine grape in North America [14]. There were also studies on the effect of shading on berry quality in the red grape variety 87-1 and the green grape variety Bixiang Seedless. Results revealed that shading the clusters with yellow paper bags with black interiors from 11:00 to 14:00 every day would slightly reduce the quality of the grape berries, but the difference was not significant. Therefore, shading treatment for a period of time can not only improve the quality of the grape berries but also affect the grape’s coloration [15]. Light quality plays an important role in grape flower bud differentiation. Previous studies have shown that the levels of indole-3-acetic acid (IAA) and abscisic acid (ABA) in grape flower buds at all stages of differentiation under red–blue light were higher than those in the control. This discovery lays the foundation for further analysis of the regulation of grape bud differentiation under red and blue light conditions [16].

The study on the effects of summer shading on the senescence of herbaceous peony leaves and the content of endogenous hormones and polyamines shows that herbaceous peonies can delay summer leaf senescence through shading and growth regulators [17]. A study was conducted to investigate the effects of shading on the growth and photosynthetic characteristics of leaves of three forage species in apple orchards on the Loess Plateau in eastern Gansu. The results showed that shading had significant effects on leaf growth and photosynthetic characteristics, but there were differences among species [18]. In the study of grape berries, the impact of shading during berry ripening on the tannin structure–activity relationship of berry skin extracts was examined, as well as the effect of vine roof shading on C6/C9 compound accumulation in the northwest grape variety ‘Cabernet Sauvignon’ (*V. vinifera* L.) [19]. The berries were the main focus of the study. The results showed that artificial shading could lead to changes in berry pigmentation and tannin composition [20].

Twenty VvLhcs have been identified in the Vitis gene family, distributed on 13 of the 19 chromosomes, divided into seven developmental branches [21]. Lhc is a nuclear gene that was discovered at the beginning of plant light collection. The earliest Lhc was identified by using mRNA synthesis of double-stranded cDNA with the rbcS gene in pea [22]. The relationship between Lhc expression and light exposure was a focus of early research. Many studies have shown that Lhc is regulated by light, and light quality regulates gene expression through pigments. Gene expression is upregulated by white and red light [23]. Photosynthesis is an important biological process in plants, supporting all forms of energy. Light harvesting is the most important step in this process, mediated by the light-harvesting chlorophyll a/b binding protein (Lhc) [24]. It is mainly located on the thylakoid membrane of chloroplasts and combines with pigments to form a light-harvesting pigment protein complex, participating in the collection and transmission of light energy during photosynthesis [25].

To ensure the stability and sustainable growth of berry quality, accumulating sugar in berries through light is the primary method to enhance fruit quality. It is necessary to implement reasonable measures to strengthen the management and protection of berries. Bagging grape berries is a physical protection method used to prevent damage to the skin of berries by blocking contact with the external environment. Bagging helps to keep the surface clean and maintain the integrity of the berries. Additionally, bagging creates various light, temperature, and thermal microenvironments, which contribute to enhancing the color and overall quality of grape berries [26]. Therefore, this study investigated the expression patterns of sugar-related genes in grape berries under various shading treatments. This was achieved by measuring the accumulation of sugar content in grape berries after shading and combining it with transcriptome analysis.

## 2. Results

### 2.1. Analysis of Grape Berry Quality after Cluster Shading

Through the analysis of the grape cluster color conversion rate (Figure 1a), it was observed that the shading effect of the D treatment and C treatment was most pronounced during the S1 period. In periods S1, S2, S4, S5, and S6, the color conversion rate of CK was higher than that of the other treatments. In the S3 period, the color conversion rate significantly increased for the four shading treatments. The preliminary results indicated that shading treatment with bags of various colors had a certain weakening effect on the color of grape clusters. During the S6 period, the cluster’s color conversion rate reached 100% under various treatments. It can be found that shading treatment can only temporarily affect the color of the grape clusters and cannot impact the final color of the clusters.

The color difference L-value of grape berry skin showed that under different treatments, the brightness of grape berries under the D treatment was significantly higher than that under the other treatments, and the overall trend varied from light to dark on different treatment days (Figure 1b). A study on grape berry color revealed that during the DAF90 period, the grape cluster color under treatments C and D tended to be green. Throughout the entire testing process, the grape cluster color under the remaining treatments tended to be red (Figure 1c). The color difference b-value of the grape cluster color indicated that the clusters treated with A, B, C, and D during the DAF90 period, and C and D during the DAF100 period, appeared yellowish. As the berries grew and developed, the grape cluster’s color gradually changed to blue. The results showed that the color of the grape cluster was affected after the shading treatment (Figure 1d).

The results showed that the single-berry weight of the grape cluster was determined at different time points and under various treatments (Figure 2a). During the DAF90 period, the weight of a single berry per CK was higher compared to other treatments. Interestingly, at DAF125, we found no significant difference in single-berry weight between the shading treatment and the control. This preliminary finding suggests that shading treatments significantly affected the weight of individual berries at early stages but did not affect their berry weight beyond 125 days after anthesis.

Analysis of the results of soluble solids in grape berries shows that, in different periods, the content of each treatment is lower than that of CK. However, it shows an upward trend throughout the entire testing process (Figure 2b).

Through the determination of soluble sugar content in berries at different time intervals and under various treatments, it can be observed (Figure 2c) that the soluble sugar content of CK was consistently higher than that of the other treatments during each time period. Additionally, the content of DFA125 was found to be 1.81 times higher than that of DFA90. Throughout the experiment, the soluble sugar content showed an upward trend.

The results of the determination and analysis of titrated acid content in berries showed that in different periods, the acidity of the berries in the shading treatment was higher than that of CK, and the D treatment exhibited the most significant difference (Figure 2d). As the grape berry ripens, the acid content shows a decreasing trend. Interestingly, at the same time, shading treatment increases the acid content in the fruit.

According to the sucrose content expression chart, the sucrose content of CK was higher than that of the other treatments at different time periods. Additionally, the content of DFA90 was 2.97 times higher than that of DFA125. During the entire experiment, the sucrose content exhibited a decreasing trend (Figure 3a). As depicted in the graph showing glucose content determination, the glucose content exhibited the most significant increase. During various time periods, the glucose content of CK was higher than that of the other treatments. Specifically, the content of DFA125 was 3.44 times higher than that of DFA90. During the entire process, the glucose content exhibited an upward trend (Figure 3b). The trend of fructose content change was similar to that of the glucose content (Figure 3c). This suggests that both glucose and sucrose increase during berry growth, and different treatments can reduce sugar accumulation. Furthermore, higher shading rates have a more pronounced effect on the sugar content of berries. 

### 2.2. Differential Gene Screening

Through transcriptome analysis of grape berries at 100 and 125 days after flowering and under various treatments (CK1, A1, B1, C1, and D1 and CK2, A2, B2, C2, and D2) [27], significant results of differential gene KEGG enrichment analysis were obtained. Table 1 shows that 16 genes were differentially expressed in the photosynthesis-antenna protein pathway, with varying levels of gene expression. 

### 2.3. Mechanism of Influence of Different Expression Groups of Proteins through the Photosynthesis-Antenna Protein Pathway on the Sugar Content in the Grape Cluster

We identified changes at the gene level by comparing data from each transcriptome, and conducted GO and KEGG enrichment analyses using all reference genes as the background to comprehend the function of DEGs. The KEGG pathways, including flavonoid biosynthesis, circadian plants, stilbene compounds, diarylheptane compound and gingerol biosynthesis, photosynthetic antenna proteins, and phenylalanine metabolism, are significantly enriched in CK1 to A1, CK1 to B1, CK1 to C1, CK1 to D1, CK2 to A2, CK2 to B2, CK2 to C2 and CK2 to D2, among which the photosynthetic antenna protein biosynthesis pathway was chosen for further analysis. Photosynthesis is the process of converting light energy into chemical energy. It involves the absorption, transfer, and conversion of light energy, electron transport, photophosphorylation, and carbon assimilation. The pigment–protein complex on the photosynthetic membrane is typically divided into four components: Photosystem II (PS II), Photosystem I (PS I), the Cytb6f complex, and the ATP synthase complex. The participation of these four protein complexes completes the electron transfer process from H2O to NADP+ in the photoreaction of photosynthesis, resulting in the production of ATP and the release of O2 [28]. From Figure 4, it is indicated that upon receiving light, the light-trapping pigments in PS I and PS II regulate the efficiency of light absorption in grape berries, which subsequently affects the sugar content of grape clusters. During the DFA100 period, we identified 10 differentially expressed genes in PS II. We observed that the gene expression levels were higher under CK1, indicating a positive regulation of the absorption efficiency of light-trapping pigment. Under treatment D1, the grape cluster was affected by shading, resulting in a decrease in gene expression levels. This, in turn, reduced the efficiency of light-capturing pigments, the decomposition efficiency of water, and the speed of entering ATP synthetase. Therefore, the efficiency of carbon fixation in plant tissues decreased, leading to a reduction in sugar content in the berries. The expression trend of VIT_12s0055g01110 (Lhcb6) was the most noticeable. In PS I, we identified six differentially expressed genes. In the control CK1 treatment, the gene expression was higher and positively regulated the light-trapping pigment. Under the treatment of B1 and C1, the expression content was low, which reduced the efficiency of light harvesting, NADPH generation, and carbon fixation, wherein the substances entering carbon fixation are reduced, leading to a decrease in sugar content in grape clusters. During the DFA125 phase, the expression of the A2 processing gene was detected in PS II, which positively regulates the absorption efficiency of light-trapping pigments. In PS I, the D2 treatment had the lowest expression level. This reduces the light-absorbing efficiency of light-trapping pigments, weakens the transfer of electrons, and decreases the efficiency of water decomposition. As a result, plants have lower carbon fixation efficiency, and berries have lower sugar content. It can be concluded that inhibiting the gene expression of the photosynthesis-antenna protein pathway in grape berries after shading the grape cluster leads to a decrease in sugar content in the berries.

### 2.4. Correlation Analysis between Berries’ Sugar Content and Differential Genes

Correlation analysis indicates that during the DFA100 period (Figure 5), apart from the three genes VIT_11s0016g00073 (Lhca2), VIT_18s0001g10550 (Lhca5), and VIT_07s0005g02220 (Lhcb1), the other 13 differentially expressed genes were positively correlated with glucose content. During the DFA125 period, only three genes, VIT_10s0003g02890 (Lhcb1), VIT_10s0003g02900 (Lhcb1), and VIT_12s0055g01110 (Lhcb6), were found to be negatively correlated with glucose content in grape berries. The remaining genes exhibited a positive correlation. The correlations of the VIT_15s0024g00040 (Lhca3) and VIT_08s0007g02190 (Lhcb4) genes with fructose, glucose, and sucrose contents were 0.71, 0.77, and 0.62, and 0.82, 0.86, and 0.79, respectively. The VIT_01s0010g03620 (Lhca2) gene was significantly correlated with sugar content. 

### 2.5. Real-Time Fluorescence Quantitative PCR Analysis

It was found from Figure 6a that the color of grape berries under the CK treatment was darker than that under the D treatment, indicating that bagging affects the color of berries at maturity. As grape berries grow, their sugar content also increases. As seen in Figure 6b, regardless of the period after flowering, the sugar content under the CK treatment is always higher than that under the D treatment. qRT-PCR tests were performed on eight genes that exhibited high expression among the identified differentially expressed genes (Figure 6c). The primer was synthesized by Shanghai Biological Company (Appendix A). The results showed that the relative expression level of CK was higher than that of the D treatment in both the DFA90 and DFA100 periods, except for VIT_19s0014g03660 (Lhcb3). The relative expression levels of VIT_12s0057g00630 (Lhcb2) and VIT_01s0010g03620 (Lhca2) in DFA90 were 1.25 times and 4 times higher, respectively, in the CK treatment compared to the D treatment. During the DFA100 period, the CK treatment was 1.61 times and 1.17 times that of the D treatment, respectively, which was consistent with the results of berry sugar content determination. The results showed that the accumulation of sugar content in grape berries could be reduced after the cluster shading treatment. 

## 3. Discussion

Improving grape berry quality has always been a focus of research both domestically and internationally. The levels of chlorophyll and anthocyanin in bagged fruits are lower than those in unbagged fruits. Nevertheless, bagging certain berries can still enhance the coloration of the fruits [29]. Grape berry quality is not only related to the size of the berries but also to their color [30]. In addition, the time of harvest also influenced the anthocyanin content of Pinot Noir grapes. Compared with the control group without defoliation, the concentration of anthocyanins in grapes at each treatment stage had a tendency to increase, and the grape color changed obviously [31]. There have also been studies on the impact of changes in inflorescence temperature on the fruit setting of Cabernet Sauvignon, and it has been found that the temperature experienced during grape flowering affects the grape setting rate and the development of seeds and grapes [32]. Shading treatment has a significant impact on the growth and development of berries. In low-light environments, berries exhibit slower growth and development, resulting in lighter coloring, especially during the grape berry expansion stage [33]. In this study, we found significant differences in grape berries between the CK and D treatments, reflected not only in the color of the berries but also in their size. In addition, the duration of light has a significant impact on the germination, growth, and soluble sugar content of grape berry leaves. Therefore, the duration of light is also a crucial factor that influences the sugar content of grape berries [34]. Although bagging guava and kiwi has been shown to make the skins scarless and pest-free, intrinsic qualities such as intrinsic sugar content have not been studied [35,36]. In comparison, this study found that in three stages, the color of berries became lighter compared to CK, and the content of glucose, sucrose, and soluble sugars significantly decreased. The sucrose content exhibited a decreasing trend throughout the entire period, with the D treatment showing a more pronounced effect. 

The reference pathways in the KEGG enrichment database can not only predict gene function but also be used to study the positions and roles of genes in different metabolic pathways [37]. A comparison of Cabernet Sauvignon berries grown in Bordeaux and Reno at similar sugar levels revealed significant differences in the expression of most transcripts at the two locations, particularly in photosynthesis and DNA metabolism [38]. Previous studies on the photosynthetic pathway and differential gene expression analysis of carrots under the condition of CO_2_ enrichment showed that the pathway with the highest concentration of differentially expressed genes was the photosynthetic pathway. These differential genes are involved in pigment synthesis, electron transport, ATP synthesis, and enzyme synthesis [39]. In addition, in a poplar heat stress response experiment, a 6 h genome-wide gene expression analysis of heat stress identified 29,898 differentially expressed genes that interact to regulate the expression of photosynthesis-related genes in response to heat stress and control carbon fixation [40]. Transcriptomic analysis of white- and red-meat apples also revealed a new gene regulating fruit color, UFGT, which is directly responsible for anthocyanin accumulation during apple ripening [41]. In this study, the KEGG enrichment results showed that there were 16 differentially expressed genes related to the expression of photosynthetic antenna protein genes in PSI and PSII. Together, they regulate the efficiency of light-trapping pigments, electron transfer, and carbon fixation. In the DFA100 and DFA125 phases, there were 16 genes regulating the photosynthetic antenna protein pathway. The expression level of VIT_12s0057g00630 (Lhcb2) was the highest. Gene expression exhibited a downward trend, which affected the efficiency of carbon sequestration and led to the decrease in sugar content in the fruit. But there are also individual genes that show the opposite trend, which may be due to the specific expression of the gene.

In higher plants, the light captured by the chlorophyll a/b-binding protein (Lhc) plays a crucial role in the plant’s response to adverse environments [21]. The emergence of Lhc is an important milestone in the evolution of photosynthetic eukaryotes. Light-harvesting chlorophyll-binding proteins form complexes near the reaction centers of PS I and II and act as antennas to collect light energy. Then, they transfer this energy to the reaction center, promoting photochemical quenching and optimizing photosynthesis [42]. In this study, a significant correlation was found between the VIT-12s0028g00320 (Lhcb1) gene and sugar content, with a correlation coefficient of 0.97, indicating that this gene plays a major role in regulating the photosynthetic pathway. Studies on Chlamydomonas reinhardtii have shown that the outer antenna system of PS I is composed of nine subunits. Compared to the antenna complexes of PS II, all Lhca complexes exhibit red-shifted fluorescence emission, similar to the Lhca complexes found in higher plants, albeit with a smaller redshift [43]. The Lhcb gene family in green plants encodes several light-harvesting Chl a/b-binding (Lhc) proteins that collect light energy and deliver it to the reaction center of PSII [44]. In this study, correlation analysis and qRT-PCR validation showed that the correlation coefficients between the VIT-12s0057g00630 (Lhcb2) and VIT-01s0010g03620 (Lhca2) genes and sugar content were 0.79 and 0.92, respectively. The trends were consistent, both leading to a reduction in the accumulation of fructose content. This result is consistent with the qRT-PCR findings, demonstrating that shading conditions can influence the expression levels of related genes, consequently diminishing carbon conversion efficiency and decreasing sugar content accumulation.

## 4. Methods

### 4.1. Plant Materials

In this study, the ‘Marselan’ grape was cultivated in the Wuwei Forestry Research Institute’s grape demonstration garden (102°42′ E, 38°02′ N), which receives abundant sunlight and experiences significant temperature variations between day and night, creating an ideal environment for grape cultivation. Grape plants with strong and consistent growth were selected, and bagging of the grape clusters was initiated at DFA 45 in the upper, middle, and lower sections. Different shading treatments were applied to the clusters using no bag (CK), a white bag (A), a yellow bag (B), a brown bag (C), and a black bag (D) with light transmission rates of 100%, 75%, 50%, 25%, and 0%, respectively. Each treatment was repeated three times. Samples were collected at DFA90, DFA100, and DFA125 to determine the quality of berries and extract berry RNA. 

### 4.2. Determination of Color Conversion Rate of Grape Berries

The color of grape clusters was analyzed at 90 days (S1), 93 days (S2), 96 days (S3), 99 days (S4), 102 days (S5), and 105 days (S6) after flowering under various treatments. Grape berries from various treatments were collected, with three clusters per treatment and three clusters per period. The grape berries were considered colored when they turned red and covered more than 50 percent of their surface area. Data on the number of colored grapes and uncolored grapes and the total grape count were gathered to determine the color conversion rate of the grape clusters. To ensure the accuracy of the test data, the average value was collected five times. 

### 4.3. Determination of Grape Berries’ Color Value

A color difference meter was used to determine the color variation of the grape clusters at various time intervals and under different treatments. The measurements were taken three times, and the color difference L-value, a-value, and b-value of the berry surface were recorded to reflect changes in the grape’s appearance color. The color difference L-value represents brightness, and the lower the L-value, the lower the brightness and the worse the glossiness of grape berries. The color difference a-value represents red versus green, with a negative value representing green. The color difference b-value indicates yellow–blue, with a positive b-value indicating yellowish and a negative b-value indicating bluish. 

### 4.4. Determination of Individual Grape Berries’ Weight

To measure the weight of single grape berries at each stage after flowering, three replicates were established. The grape berries were placed on an electronic scale with an accuracy of 0.001 g (METTLER TOLEDO, Inc., Zurich, Switzerland) to measure their weight and collect data. 

### 4.5. Determination of Soluble Solids and Soluble Sugar in Grape Berries

The soluble solid content was determined using a hand-held refractometer (ATAGO CO, LTD., Tokyo, Japan), and the soluble sugar content was determined by referring to Zhang et al. [45].

### 4.6. Determination of Acid Content in Grape Berries

The titratable acidity of grape berries was measured using a reagent kit (Shanghai Saint-Bio Biotechnology Co., Ltd., Suzhou, China). We collected 5 g of grape berries from different periods and stages and followed the instructions of the reagent kit. 

### 4.7. Determination of Sugar Components in Grape Berries

The carbohydrate components and contents of the berries were determined using a high-performance liquid chromatograph (American Waters Acquity Arc, Milford, MA, USA), following the method described by Bernardez et al. [46].

(1) Methodology for sample extraction: After cryogenically grinding the grape flesh, 0.5 g of the sample was precisely weighed and transferred to a 10 mL centrifuge tube. Subsequently, 5 mL of 80% ethanol was added, followed by ultrasonic extraction at a temperature of 35 °C for a duration of 20 min. The resulting mixture was then subjected to centrifugation at a speed of 12,000 revolutions per minute for a period of 15 min in order to obtain the supernatant. The extraction process was repeated twice, with the addition of 2 mL of 80% ethanol each time. The supernatant was combined and adjusted to a final volume of 10 mL. Subsequently, 2 mL of the resulting supernatant was evaporated at 60 °C in a vacuum centrifuge concentrator for 3 h until completely dry. It was then reconstituted with 1 mL of ultra-pure water and 1 mL of acetonitrile, followed by filtration through a microporous filter membrane with a pore size of 0.22 μm to obtain the organic-phase filtrate. Finally, three parallel tests were conducted for each sample variety.

(2) Chromatographic condition: an XBridge BEH Amide column (4.6 × 150 mm, 2.5 μm) was used with 75% acetonitrile +0.2% triethylamine +24.8% ultra-pure water as the mobile phase. The flow rate was 0.8 mL/min, the detection wavelength was 254 nm, the column temperature was 40 °C, the sample size was 10 μL, the mobile phase was filtered by a 0.22 μm organic filter membrane before use, and ultrasonic degassing was performed.

### 4.8. Correlation Analysis of Sugar Content in Grape Berries and Differential Genes

Correlation analysis can reveal the correlation degree of two correlated variable elements. In order to explore whether the sugar content in berries is related to differential genes, by measuring the quality of the berries, the levels of fructose, glucose, and sucrose in grape berries were obtained under various treatments and at different time intervals. Citing the results of a transcriptome analysis [27], it is necessary to identify the DEGs of antenna proteins through Kyoto Encyclopedia of Genes and Genomes (KEGG) enrichment analysis. To further explore the relationship between genes and sugar content, correlation analysis was conducted using SPSS software. Additionally, the ChiPlot website (https://www.chiplot.online/) (accessed on 5 February 2024) was utilized to create visual representations.

### 4.9. RNA Extraction and qRT-PCR

Three grape berries were collected from plants treated with D and CK, specifically DFA90, DFA100, and DFA125. Total RNA was extracted using the Real Times Biotechnology Beijing kit in China and verified by running it on 1% agarose gel stained with GoldView. Eight differentially expressed genes that show a clear trend in the photosynthesis-antenna protein pathway and have a strong correlation with sugar content were chosen for real-time fluorescence quantitative PCR. The test primers were synthesized by Shanghai Bioengineering, and the specific primers are shown in Table 1. All data are expressed as averages ± SE. The significance level is defined as *t* test 0.01 ≤ *p* ≤ 0.05 (*), 0.001 ≤ *p* ≤ 0.01 (**), and *p* ≤ 0.001 (***), while one-way analysis of variance (ANOVA) represents *p* < 0.05. 

### 4.10. Data Analysis

The collected data were sorted, classified, and analyzed. Microsoft Excel 2016 was used for statistical processing, while IBM SPSS Statistics 26 software was utilized for data variance analysis and significance analysis. Origin 2023 was used for drawing.

## 5. Conclusions

Research data show that shading can reduce berry color conversion, grape color, sugar content, single-grain weight, and soluble solid content, as well as berry efficiency. Additionally, varying levels of light transmittance can also impact the accumulation of sugar content in grape berries. Transcriptomic analysis revealed 16 differentially expressed genes regulating PS I and PS II. These genes influence the accumulation of sugar content in the berries through the photosynthetic pathway. The results showed that the gene expression of the photosynthetic antenna protein pathway in berries was inhibited after cluster shading. This inhibition may lead to a reduction in the sugar content of grape berries.

## Figures and Tables

**Figure 1 ijms-25-05029-f001:**
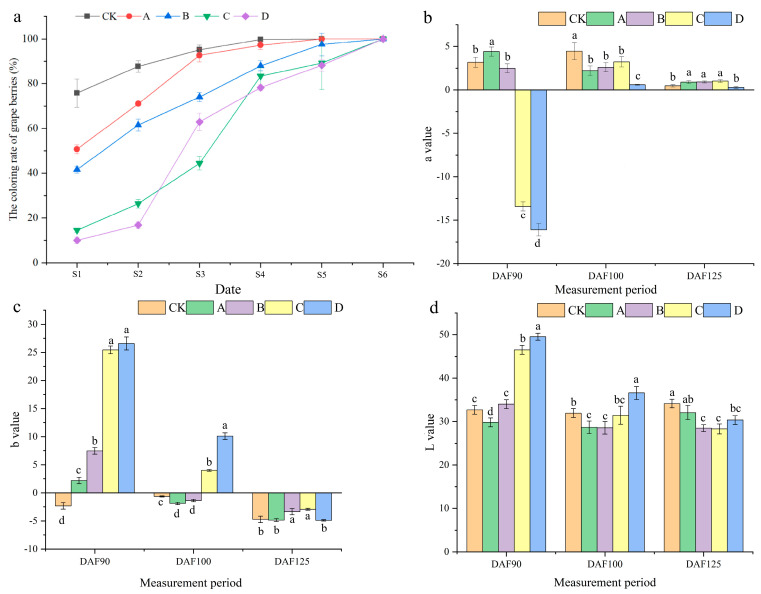
Determination of external quality of grape berries at different periods. (**a**) Color conversion rate of grape berries in different periods. (**b**) a-value of grape berries in three periods. (**c**) b-value of grape berries in three periods. (**d**) L-value of grape berries in three periods. The data are expressed as average values, and different letters indicate significant differences between treatments (*p* < 0.05).

**Figure 2 ijms-25-05029-f002:**
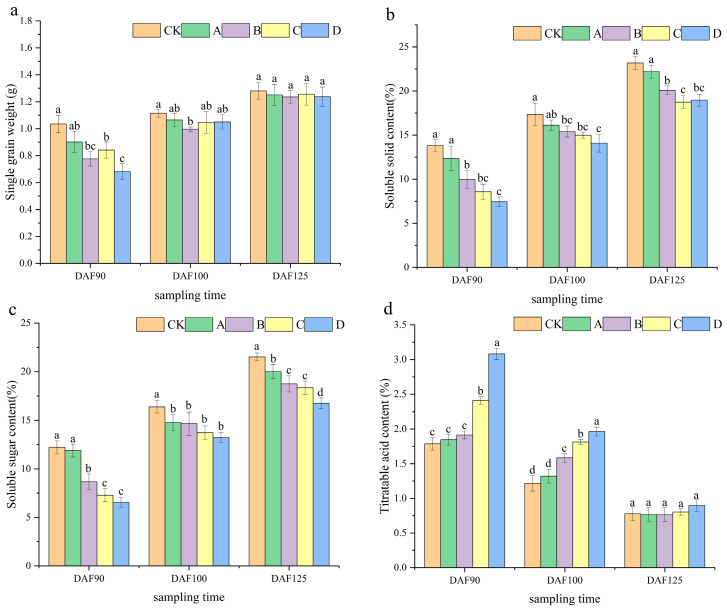
Determination of internal quality of grape berries at different periods. (**a**) Single-grain weight of grape berries in three periods. (**b**) Soluble solid content of grape berries in three periods. (**c**) Soluble sugar content of grape berries in three periods. (**d**) Titratable acid content of grape berries in three periods. The data are expressed as average values, and different letters indicate significant differences between treatments (*p* < 0.05).

**Figure 3 ijms-25-05029-f003:**
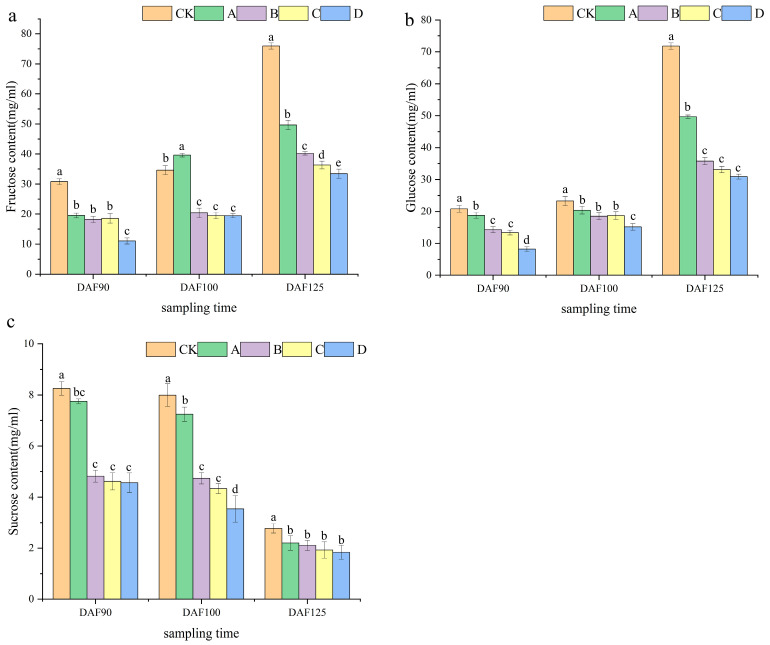
Determination of sugar content in grape berries at different periods. (**a**) Fructose content of grape berries in three periods. (**b**) Glucose content of grape berries in three periods. (**c**) Sucrose content of grape berries in three periods. Data are expressed as averages. Different letters indicate significant difference between treatments (*p* < 0.05).

**Figure 4 ijms-25-05029-f004:**
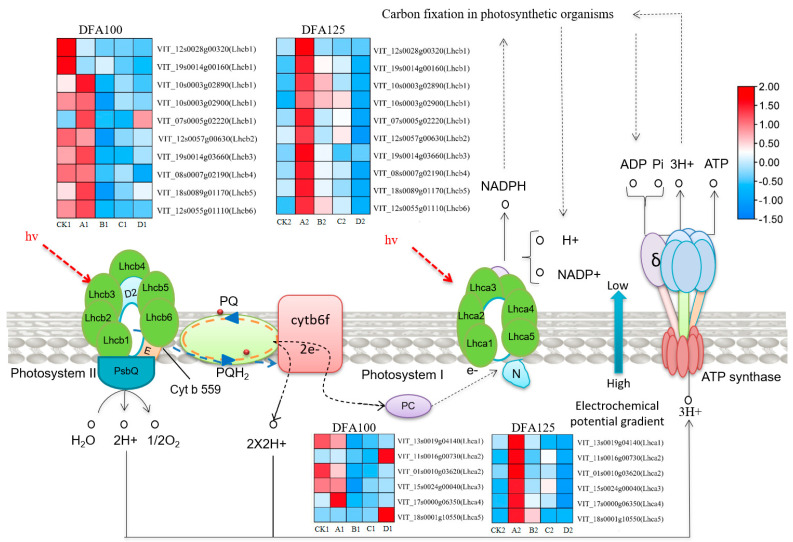
The contents of 16 differential genes in the photosynthesis-antenna protein pathway and their regulatory roles in plant PS I and PS II.

**Figure 5 ijms-25-05029-f005:**
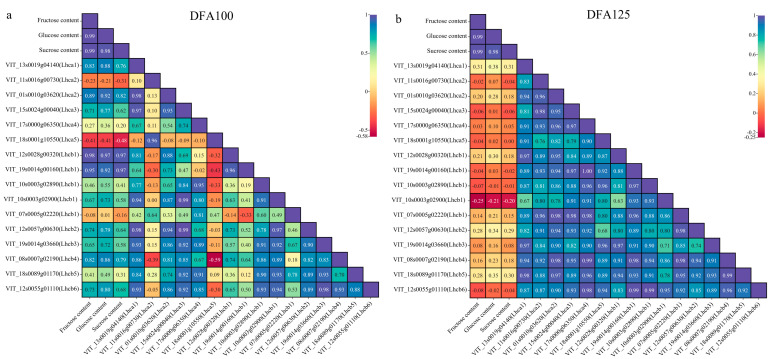
Correlation analysis of glucose, fructose, and sucrose content and 16 differential genes in grape berries. (**a**) Correlation between gene expression and berry sugar content at 100 days after flowering. (**b**) Correlation between gene expression and berry sugar content at 125 days after flowering.

**Figure 6 ijms-25-05029-f006:**
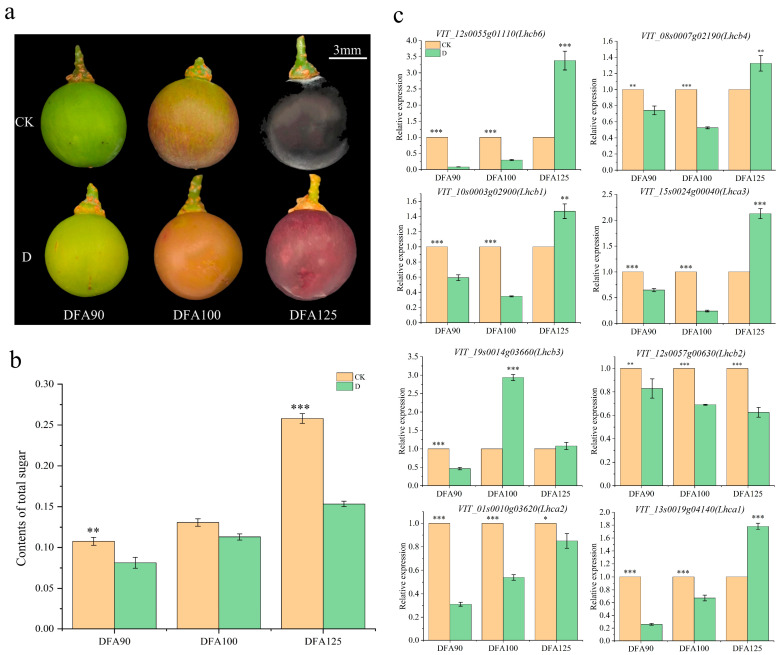
Pictures of (**a**) CK (control)- and D (shading)-treated grape berries at different periods. (**b**) Total sugar content of berries in the three periods. (**c**) The relative expression levels of 8 differentially expressed genes determined by real-time fluorescence quantification. All data are expressed as average ± SE. The significance level is defined as *t* test 0.01 ≤ *p* ≤ 0.05 (*), 0.001 ≤ *p* ≤ 0.01 (**), and *p* ≤ 0.001 (***), while one-way analysis of variance (ANOVA) represents *p* < 0.05.

**Table 1 ijms-25-05029-t001:** The expression of 16 differentially expressed genes in the photosynthetic antenna protein pathway.

Gene Name	CK1	A1	B1	C1	D1	CK2	A2	B2	C2	D2
VIT_12s0028g00320(Lhcb1)	22.44	5.17	1.28	1.38	0.32	31.77	198.3	24.05	5.99	0.51
VIT_19s0014g00160 (Lhcb1)	6.37	1.31	2.07	0.67	0.11	24.2	152.1	75.2	61.23	4.57
VIT_10s0003g02890 (Lhcb1)	45.46	82.69	9.38	27.24	16.56	12.64	81.25	55.62	27.65	2.48
VIT_10s0003g02900 (Lhcb1)	2.56	2.77	0.66	1.58	1.43	0.19	2.26	1.72	1.53	0.001
VIT_07s0005g02220 (Lhcb1)	0.65	1.84	0.45	0.34	1.49	2.32	6.92	2.79	3.54	0.2
VIT_12s0057g00630 (Lhcb2)	288.1	271.81	166	213.16	225.26	260.39	498.7	230.66	320.38	29.48
VIT_19s0014g03660 (Lhcb3)	5.24	6.62	1.01	1.16	2.68	0.99	6.54	2.41	0.46	0.75
VIT_08s0007g02190 (Lhcb4)	198.8	195.84	89.87	116.97	64.33	63.1	185.61	89.35	76.43	9.02
VIT_18s0089g01170 (Lhcb5)	91.56	123.35	34.53	63.58	81.01	25.3	63.52	29.35	20.13	2.86
VIT_12s0055g01110 (Lhcb6)	9.04	10.49	2.94	3.46	3.81	3.58	16.64	10.63	7.4	1.73
VIT_13s0019g04140 (Lhca1)	46.18	38.13	4.45	12.18	18.74	12.16	38.17	13.26	3.18	0.31
VIT_11s0016g00730 (Lhca2)	1.69	1.99	0.82	0.26	5.98	0.15	1.51	0.24	0.67	0.001
VIT_01s0010g03620 (Lhca2)	100.03	77.35	39.86	45.95	59.26	36.47	113.92	35.2	44.08	6.01
VIT_15s0024g00040 (Lhca3)	85.94	84.83	52.12	65.8	67.72	13.58	63.26	25.95	37.01	5.42
VIT_17s0000g06350 (Lhca4)	12.69	30.96	4.53	7.19	10.68	3.95	19.12	9.16	7.82	0.08
VIT_18s0001g10550 (Lhca5)	0.22	0.22	0.08	0.19	2.4	0.001	0.16	0.09	0.001	0.001

## Data Availability

Data available on request from the authors. The data that support the findings of this study are available from the corresponding author upon reasonable request.

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
