# Peer review of "Shading Treatment Reduces Grape Sugar Content by Suppressing Photosynthesis-Antenna Protein Pathway Gene Expression in Grape Berries"

_ijms, 2024, doi:10.3390/ijms25095029_

Round 1
Reviewer 1 Report
Comments and Suggestions for Authors
In this research manuscript, the authors Nan et al have investigated the effect on grape cluster shading on the soluble sugar content, acid content, soluble solid content of the grapes. The gene expression analysis and transcriptomic analysis further help in understanding the differently expressed genes that influence the quality of grapes harvested.
Minor comments include:
1) Please provide references for line 33 in the introduction section- “Grapes are rich in nutrient…human health.”
2) Line 40, add full stop after reference [6-7].
3) The introduction lists the different sugars found in grape berries. Similarly it will be informative to mention the names of the acids found in grape berries in line 41 : “The composition and proportion of sugar and acid in …… other components”.
4) The labels and axis numbers look hazy for figure 1a, b, c,d, figure 2a, b, c,d, figure 3 a, b,c and figure 6b and 6c look hazy. The resolution of the figures can be improved.
5) Similarly, the gene names in figure 4, 5 are not legible. Please provide a better quality of images.
6) In results section 2.1: line 129 “The results… showed that (Figure 2d). In different periods………” Combine the two sentences to one. The revised sentence can be “The results of the determination and analysis of titrated acid content in berries showed that in different periods, the acidity of the berries in the shading treatment was higher than that of the CK, and the D treatment exhibiting the most significant difference (Figure 2d).”
7) In section 3, Discussion, line 269, “In this study, KEGG….. protein genes in PSI and PSI.” Did the authors mean PSI and PSII?
Author Response
Thank you for your review,please see the attachment.

Reviewer 2 Report
Comments and Suggestions for Authors
Dear Authors,
This study investigates the effect of shading 'Marselan' grape berries under different light transmission rates (100% (CK), 75% (A), 50% (B), 25% (C), 0% (D)) on their ripening process It was found that shading delayed the ripening of the grapes. The individual weight of the grapes, along with the content of fructose, glucose, soluble sugars, and organic acids, was measured at different days after flowering (DAF90, DAF100, DAF125).
For a comprehensive review, the manuscript requires major revisions. Detailed comments and critiques have been included throughout the manuscript, highlighting areas needing extensive elaboration, particularly in the results and discussion sections. The manuscript must thoroughly detail these sections to provide a clear, comprehensive understanding of the findings. Additionally, due to image quality issues, the figures are unreadable and cannot be followed. A significant improvement in figure presentation and clarity is necessary for effective communication of the results. The extensive comments marked in the manuscript should guide the revision process.
Please review my attached comments on the manuscript.

Comments on the Quality of English LanguageModerate editing of English language required.
Author Response

(The authors gave the same response as above.)

Reviewer 3 Report
Comments and Suggestions for Authors
The article “Shading Treatment Reduces Grape Sugar Content by Suppressing Photosynthesis-Antenna Proteins Pathway Gene Expression In Grape Berries” is written in simple and understandable language. With the help of transcriptome analysis, they were able to prove a simple and well-known fact, namely that the quality and quantity of light has a direct influence on the color of grape berries and the sugar content.
Pay attention to the order in which you have written the tables: You wrote table 2 first and then table 1. For better understanding, explain the abbreviations (C1 D1 CK2 A2 B2 C2 D2 CK1 A1 B1) you use in the table under Table 2, which should actually be Table 1. In general, follow the instructions for writing an article.
Considering that many similar research papers have been published, the Discussion should better be written using additional references, such as Song et al. BMC Plant Biology 2014, 14:111 http://www.biomedcentral.com/1471-2229/14/111
Author Response

(The authors gave the same response as above.)

Reviewer 4 Report
Comments and Suggestions for Authors
The manuscript presents the impact of shading treatment on grape berries and the subsequent reduction of sugar content through the suppression of gene expression in the Photosynthesis-Antenna Proteins pathway. However, the findings lack novelty as the effect of shade treatment on photosystem reaction centers is well-established. To enhance the novelty, the authors should explore alterations in secondary metabolites and identify novel pathways modulated under shade conditions to alter acidity.
1. The manuscript demonstrates a bias towards studies conducted in a specific country. It is imperative to include references from grape quality assessments conducted worldwide to provide a more comprehensive perspective.
2. In the abstract, the statement regarding sugar content changes under shade treatment is ambiguous. Since shade treatment typically reduces sugar content, clarity is needed to ensure the findings are accurately communicated.
3. A global report on the grape industry should be cited to provide a broader context for the study.
4. The statement regarding consumer preferences for grape quality needs clarification. The authors should elucidate what aspects of grape quality consumers prioritize in modern times.
5. It is advisable to introduce and explain color parameters such as L and b values before discussing their relevance to the study results.
6. Clarity is needed regarding whether the authors refer to a reduction in acid content compared to the Day After Flowering (DAF) or shade treatment in line 132-133.
7. The section on "differential gene screening" appears disjointed. The rationale for selecting only antenna proteins and the total list of proteins should be provided. Additionally, more information supporting previous publications on this topic would enhance the section.
8. The correlation studies between sugar content and Differentially Expressed Genes (DEGs) lack contextual explanation. The authors should clarify the rationale behind these studies.
9. The primer sequences in Table 1 could be moved to supplementary data to streamline the main manuscript.
10. Avoid phrases like "my research" and maintain a third-person perspective throughout the manuscript, as seen in line 254-255.
11. While the role of light in activating photosynthesis and sugar anabolism is well-established, the manuscript should explore other metabolic pathways affecting acidity or pigment content in berries. Additionally, measuring secondary metabolite levels under different shade treatments would add depth to the study.
Comments on the Quality of English LanguageSome of the sentences are out of context in the manuscript and must be thoroughly revised.
Author Response

(The authors gave the same response as above.)

Round 2
Reviewer 2 Report
Comments and Suggestions for Authors
Dear Authors,
I thank the authors for improving the figure quality of the manuscript. In this case, the manuscript became quite understandable and the necessary changes to the article were completed in line with my suggestions. I therefore confirm that the manuscript is acceptable for publication.
Comments on the Quality of English LanguageMinor editing of English language required.
Author Response
We have revised your suggestion, please check in the attachment.

Reviewer 4 Report
Comments and Suggestions for Authors
The manuscript looks better than the previous version but some of the major issues still remain which have been reiterated as follows:
Grape as claimed is a widely grown plant and its importance is enormous. To be useful to the broad readership inclusion of more articles are important. List of some of the publications are as follows:
1. Timing of leaf removal modulates tannin composition and the level of anthocyanins and methoxypyrazines in Pinot noir grapes and wines
2. A sense of place: transcriptomics identifies environmental signatures in Cabernet Sauvignon berry skins in the late stages of ripening
3. Berry skin development in Norton grape: distinct patterns of transcriptional regulation and flavonoid biosynthesis
4. Inflorescence temperature influences fruit set, phenology, and sink strength of Cabernet Sauvignon grape berries
I could not find the transcriptomic data deposited in a public repository along with the relevant details related to the RNAseq study.
Regarding point 7 of my previous comments, it is not clear what are the enrichment plots refer to. What are the DEG drawn across. The first and the third plot looks the same. All this must be included ion the manuscript with a clear explaination of the rationale behind selecting the antennae proteins.
Point 11, I dont get what breakthrough the authors refer to include the data about other secondary metabolic pathways.
Author Response

(The authors gave the same response as above.)
